# A Novel Pyroptosis-Related Gene Signature for Prediction of Disease-Free Survival in Papillary Thyroid Carcinoma

**DOI:** 10.3390/jpm13010085

**Published:** 2022-12-29

**Authors:** Kecheng Jiang, Bei Lin, Yu Zhang, Kaining Lu, Fan Wu, Dingcun Luo

**Affiliations:** 1The Fourth Clinical Medical College, Zhejiang Chinese Medical University, Hangzhou 310053, China; 2Department of Surgical Oncology, Affiliated Hangzhou First People’s Hospital, Zhejiang University School of Medicine, Hangzhou 310006, China

**Keywords:** papillary thyroid carcinoma, pyroptosis, disease-free survival, tumor microenvironment, gene signature

## Abstract

Background: The incidence and recurrence rate of papillary thyroid carcinoma (PTC) are high. Thus, it is critical to accurately identify patients at high risk of recurrence. Pyroptosis is a type of programmed cell death closely related to the progression and prognosis of cancer. However, the role of pyroptosis in PTC remains unclear. Methods: Transcriptome data for PTC patients were obtained from The Cancer Genome Atlas database. The expression level of pyroptosis-related genes (PRGs) in PTC and normal tissues was identified. Based on these differentially expressed genes, a risk score model of disease-free survival (DFS) was established using least absolute shrinkage and selection operator Cox regression. In-cluster and quantitative real-time PCR validations were carried out. A nomogram, in combination with clinical factors, was also established. In addition, its relationship with immune characteristics and tumor gene mutations is discussed. Results: A risk score model with four PRGs, including *CASP6*, *CASP9*, *IL-18*, and *NOD1*, was established. The samples were divided into high- and low-risk clusters, according to the risk score, revealing significant differences in DFS between the two clusters. A nomogram was established combining age, lymph node metastasis and extrathyroidal extension. The area under the curve (AUC) of predicting one-, five-, and 10-year DFS in PTC patients was 0.745, 0.801, and 0.803, respectively. The low-risk cluster showed higher levels of immune infiltration and immune checkpoint gene expression, while the high-risk cluster demonstrated a higher tumor mutation burden. Conclusion: A predictive DFS model was established, based on PRGs, which may aid in identifying patients at high risk of recurrence. The present study helps to better understand the role of pyroptosis in the progression and prognosis of PTC.

## 1. Introduction

Thyroid cancer is the most common malignant tumor of the endocrine system, and its incidence has been increasing over the past decade [1]. Papillary thyroid carcinoma (PTC) is the most common subtype of all thyroid carcinomas. It is generally characterized by a high degree of differentiation, slow clinical course, and low specific mortality rate. However, a considerable number of patients still experience persistent or recurrent disease, with an incidence of 14–30% [2,3,4]. Stratification systems currently recommended by the American Thyroid Association, and European Thyroid Association, are organized mainly based on the aspects of primary tumor size, lymph node, and pathology. Unfortunately, about 10% of patients classified as low-risk still experience progression or recurrence [5]. For PTC patients, regular postoperative review and thyroid-stimulating hormone (TSH) suppression should be performed to reduce the risk of tumor recurrence [6]. However, long-term TSH suppression therapy also brings about a series of adverse reactions, such as osteoporosis, arrhythmia, and hyperthyroidism [7]. Therefore, a more accurate assessment of the recurrence risk is essential to avoid overtreatment in low-risk patients and to ensure that high-risk patients receive more aggressive treatment, such as higher TSH suppression and more frequent postoperative follow-ups.

Programmed cell death, including apoptosis, autophagy, and pyroptosis, refers to the way in which cells die depending on the signal or activity encoded by specific genes. It is a special mechanism regulating cell proliferation, homeostasis, and tumor development [8]. Pyroptosis is a type of programmed cell death mediated by the gasdermin protein family. It is triggered by a specific inflammasome, leading to the activation of the canonical caspase-1 or the non-canonical caspase-4/5/11 inflammasome pathways. Caspase can cleave gasdermin D and form plasma membrane pores. Eventually, this results in cytoplasmic swelling, membrane rupture, and release of inflammatory factors, inducing an inflammatory response [9,10,11]. Pyroptosis was initially thought to play a key role in fighting infection, but related studies have shown that pyroptosis is closely related to cancer. For example, the programmed death ligand 1 (PD-L1) has been found to switch cell apoptosis into pyroptosis, leading to tumor necrosis [12]. It has also been reported that chemotherapy drugs can induce cell pyroptosis via caspase-3 cleavage of gasdermin E [13]. In general, the impact of pyroptosis on cancer may be two-sided. On one hand, pyroptosis can eliminate cancer cells and inhibit tumor growth. On the other hand, pyroptosis may also form a suitable microenvironment for tumor cell growth [14]. Several studies based on public databases have investigated the value of pyroptosis in predicting cancer prognosis, such as in gastric [15] and ovarian cancers [16]. The mechanism and potential role of pyroptosis in tumor regulation can provide new strategies for cancer treatment.

At present, there are still few studies on the relationship between pyroptosis and thyroid cancer. Previous studies [17] reported the role of PRGs in predicting the overall survival of PTC. However, in view of the extremely low mortality rate of PTC, it is more clinically meaningful to focus on the recurrence and progression of PTC. The present study determined the expression level of pyroptosis-related genes (PRGs) in PTC, explored the value of these genes in predicting disease-free survival (DFS), established a nomogram, and discussed its relationship with immune characteristics and tumor mutations. In addition, the expression of PRGs was verified using quantitative real-time PCR in clinical tissue samples.

## 2. Materials and Methods

### 2.1. Datasets and PRGs

The normalized RNA expression, simple nucleotide variation, and clinical data were all obtained from The Cancer Genome Atlas (TCGA) database. The clinical data included age, sex, lymph node metastasis (LNM), extrathyroidal extension (ETE), and BRAF mutation. Sample inclusion criteria were as follows: 1. Pathological type of PTC; and 2. No missing DFS data. Overall, 502 PTC and 58 normal samples were included in the study. A total of 33 PRGs were obtained from previous reviews [18,19,20,21] (Appendix A).

### 2.2. Identification of Differentially Expressed PRGs

The “limma” package [22] was used to identify PRGs differentially expressed in PTC and normal tissues. The “pheatmap” package (https://CRAN.R-project.org/package=pheatmap, accessed on 1 May 2022) was used to show the expression of these differential PRGs. Then, the Search Tool for Retrieval of Interaction Genes was used to create a protein-protein interaction (PPI) network map. In addition, the “corrplot” package (https://github.com/taiyun/corrplot, accessed on 1 May 2022) was used to evaluate the correlation between PRGs via a Spearman’s rank correlation.

### 2.3. Establishment of Risk Score Model Based on PRGs

The tumor samples were randomly divided into two different subgroups, such that 70% of patients were included in the training set and 30% in the validation set. In the training set, PRGs associated with DFS were screened using univariate Cox regression, where the *p* value of <0.2 was set as the risk score formula as follows (exp: gene expression level, *β*: coefficients):$$risk\;score=\overset{n}{\underset{i=1}{\sum }}\exp i*\beta i$$

Then, the samples were divided into low- and high-risk clusters based on the median risk score, and the results were visualized using the “Rtsne” package [23]. The principal component analysis (PCA) and t-distributed stochastic neighbor embedding (t-SNE) analyses were used to identify different distributions between the two risk clusters. Kaplan–Meier survival curves were generated using the “survival” and “survminer” packages (https://CRAN.R-project.org/package=survminer, accessed on 1 May 2022) to compare the DFS differences between the two clusters. The time–receiver operating characteristic (ROC) analysis was performed using the “timeROC” packages [24] to evaluate the predictive value of the risk score in the training, testing, and total data sets.

### 2.4. Immune Characteristics Analysis

The CIBERSORT algorithm (http://cibersort.stanford.edu/, accessed on 1 May 2022) was used to calculate the fractions of infiltrating immune cells. The ESTIMATE algorithm (https://bioinformatics.mdanderson.org/estimate/, accessed on 1 May 2022) was used to calculate the StromalScore, ImmuneScore, and ESTIMATEScore. The student T test and Hotelling’s T2 test were utilized to compare the differences in immune characteristics between the two clusters. Then we analyzed the correlation between the infiltration degree of 22 kinds of immune cells and the risk score. In addition, the expression of immune checkpoints was compared between the two clusters.

### 2.5. Gene Mutation Analysis

The “maftools” package [25] was used to visualize the gene mutation profile of the high- and low-risk clusters. The gene mutation rate was then compared between the high- and low-risk clusters. TMB (tumor mutation burden) is defined as the total number of non-synonymous somatic mutations identified per megabase in the tumor genome [26]. This parameter was calculated using simple nucleotide variation data.

### 2.6. Functional Enrichment Analysis

To further investigate the differences in gene function and the pathway between high- and low-risk clusters, the “limma” package was used to screen differentially expressed genes (DEGs), based on the False Discovery Rate (FDR) < 0.05 and |log2FC| ≥ 1 standards. Then, the “ClusterProfiler” package [27] was used for gene ontology (GO) and Kyoto Encyclopedia of Genes and Genomes (KEGG) enrichment analyses.

### 2.7. Nomogram Establishment

The risk scores were further combined with clinical factors to establish predictive DFS models and a nomogram. The clinical factors were screened via Lasso Cox regression. The prognostic model was then established and the nomogram was plotted using the “regplot” package (https://CRAN.R-project.org/package=regplot, accessed on 1 May 2022). The “TimeROC” package was used for ROC curve analysis for one, five, and ten years to evaluate the prediction efficiency of the nomogram.

### 2.8. The qRT–PCR and Human Protein Atlas (HPA) Database Validation

A total of 30 pairs of tumor and adjacent normal tissue samples were collected in this study. All samples were from patients who underwent surgical resection in the Department of Surgical Oncology of Hangzhou First People’s Hospital. Postoperative pathology results confirmed the PTC diagnosis. The total RNA samples were extracted using TRIzol reagent (Invitrogen, Carlsbad, CA, USA). Then, the total RNA was reverse-transcribed into cDNA using HiScript II Q RT SuperMix for qPCR (+gDNA wiper) (Vazyme, Nanjing, China). Quantitative real-time PCR was performed with cDNA using ChamQ Universal SYBR qPCR Master Mix (Vazyme). GAPDH was used as an internal control for normalization. The relative fold changes in expression were calculated. All primer sequences are listed in the Appendix A. For the qRT–PCR results, we used the paired T-test to compare tumor and adjacent tissue. In addition, the protein expression of PRGs was examined using the HPA database.

### 2.9. Statistical Analysis

R software (version 4.1.2, The R Foundation for Statistical Computing, Vienna, Austria) and GraphPad Prism (version 8.3.1, GraphPad Software. San Diego, CA, USA) were used for all data analysis in this study. Unless otherwise specified, bilateral *p* < 0.05 served as the significance threshold.

## 3. Results

### 3.1. PRG Expression in Tumor and Normal Samples

First, the expression of 33 PRGs in tumor and normal samples was analyzed and compared. The results showed that the expression of 22 PRGs was different between the two groups (Figure 1A). An association network diagram was then constructed in order to further explore the interaction and correlation of PRGs (Figure 1B). The PPI analysis of these PRGs was also conducted (Figure 1C). The minimum required interaction score was set to 0.4. The PPI network contained 22 nodes and 90 edges. Cytoscape software (version 3.5.1, Cytoscape Consortium, San Diego, CA, USA) was used to identify the hub genes. The top five hub genes were *TNF, IL18, CASP1, PYCARD,* and *NLRC4* (Figure 1D).

### 3.2. Establishment of Risk Score Model Based on PRGs

First, univariate Cox regression analysis of the training set was conducted on the 22 PRGs and reserved those conforming to the criteria of *p* < 0.2. As a result, seven DFS-related PRGs (Figure 2A) were screened out, including *CASP6, CASP9, IL18, IL1B, NLRC4, NOD1*, and *PYCARD*. Then, four genes were preserved via Lasso Cox regression analysis (Figure 2B,C), and a DFS risk score model was constructed, according to the optimal λ value. The risk score was calculated as follows: risk score = (0.218 × *CASP6* exp.) + (−0.205 × *CASP9* exp.) + (−0.123 × *IL18* exp.) + (−0.037 × *NOD1* exp.) Then, the samples in the training, test, and total sets were divided into high- and low-risk clusters according to the median risk score (Appendix A). PCA and T-SNE analyses in the training sets showed that this model could distinguish patients at different risk levels (Figure 3A,B). Kaplan–Meier analysis showed that DFS was significantly lower in the high-risk cluster than in the low-risk cluster (Figure 3C). The area under the curve (AUC) for DFS, predicted using risk scores, was 0.677 at three years, 0.688 at five years, and 0.721 at 10 years (Figure 3D). Additionally, the patient survival status and risk score control are shown in Figure 3F. The predictive power of the risk score was then verified on the test sets and total set. In the test set, PCA and T-SNE showed that patients at different risks could be distinguished (Figure 4A,B), and there were significant differences in DFS between high- and low-risk clusters (Figure 4C). The AUCs for one-, five-, and ten-year DFS were 0.596, 0.622, and 0.713, respectively (Figure 4D). The validation results in the total set were similar to the above results, and there were significant differences in DFS between the two risk clusters (Figure 5C). The AUCs for one-, five-, and ten-year DFS were 0.660, 0.660, and 0.715, respectively (Figure 5D).

### 3.3. Functional Enrichment Analysis

A total of 177 DEGs were identified between the two clusters. Compared to the low-risk cluster, 68 genes were up-regulated and 109 genes were down-regulated in the high-risk cluster (Appendix A). GO enrichment and KEGG pathway analyses were conducted based on these DEGs. The results showed that DEGs play a major role in immune cell differentiation, immune response, and receptor interaction (Figure 6A,B).

### 3.4. Immune Characteristics Analysis

The differences among 22 types of immune cells, StromalScore, ImmuneScore, and ESTIMATEScore were compared between the high- and low-risk clusters. The Hotelling’s T2 test (*p* < 0.01) indicated that the level of immune cell infiltration in the low-risk group was higher than that in the high-risk group (Figure 7A), especially for naive B cells, gamma delta T cells, M0 macrophages, resting dendritic cells, and eosinophils. In the high-risk cluster, infiltration levels of CD8T cells, resting Natural killer (NK) cells, activated NK cells, and monocytes were higher. Similarly, the ESTIMATE score also showed a higher degree of immune infiltration in the low-risk cluster (*p* = 0.011) (Figure 7B). Further analysis of the correlation between risk scores and levels of immune cell infiltration showed that the activated NK cells (R = 0.18, *p* = 0.012) and activated dendritic cells (R = 0.20, *p* = 0.049) were positively correlated with the risk score (Figure 7C). However, naive B cells (R = −0.14, *p* = 0.044), M1 macrophages (R = −0.15, *p* = 0.037), and gamma delta T cells (R = −0.23, *p* = 0.011) were negatively correlated with the risk score. In addition, the comparison of immune checkpoint expression levels between the high- and low-risk clusters suggested that the low-risk cluster had higher overall expression levels (*p* < 0.01) (Figure 7D).

### 3.5. Gene Mutation Analysis

The differences in the distribution of cell mutations between high- and low-risk clusters were compared first. The top 20 genes with the most frequent mutations are shown in Figure 8A, where the mutation rates of BRAF and NRAS are significantly different between the two clusters. Evaluation of the relationship between the risk score and TMB showed that there was a significant correlation between TMB level and risk score (Figure 8B,C).

### 3.6. Nomogram Establishment

A nomogram was established by combining the risk score with clinical factors. The baseline clinical factors are shown in Table 1. Four factors were retained via Lasso Cox regression analysis, including risk score, age, LNM, and ETE (Figure 9A). Results of a multivariate Cox regression analysis for these four factors (Appendix A) was represented by a nomogram (Figure 9B). The nomogram AUC values used to predict one-, five-, and ten-year DFS in PTC patients were 0.761, 0.801, and 0.803, respectively (Figure 9C), indicating a good predictive value.

### 3.7. Gene Expression Levels

Then, qRT-PCR and HPA databases were used to verify the expression levels of the four PRGs in the risk score model. The qRT–PCR results showed that there were significant differences in the expression levels of all four genes in the samples, with *CASP6* and *NOD1* up-regulated and *CASP9* and *IL18* significantly down-regulated in tumors. The expression of *CASP6, CASP9*, and *NOD1* was consistent with the results of the bioinformatics analysis, while the expression of IL18 was the opposite of those results. The protein expression levels of the prognostic PRGs were validated using data from the HPA database. Protein expression of three PRGs was then determined, where *CASP6* and *IL18* were increased in thyroid cancer tissues, while *CASP9* was down-regulated, which was also consistent with the bioassay results (Figure 10).

## 4. Discussion

The incidence of thyroid cancer is increasing worldwide. Although surgery is an effective treatment, there is still a risk of recurrence, which introduces significant challenges to clinical treatment [3]. Therefore, it is necessary to accurately predict the recurrence of thyroid cancer. Pyroptosis has been confirmed to be associated with the development of many cancers [21]. However, its role and mechanism in PTC remain to be studied. Therefore, the present study aimed to explore the value of pyroptosis in predicting recurrence and treatment. We first obtained the transcriptome data of 510 PTC samples and 58 normal tissues from the TCGA database, and obtained 33 PRGs from the literature. Then we detected the expression of PRGs in PTC and normal tissues. Based on these differentially expressed genes, the risk score models of DFS were established, based on four PRGs, including *CASP6, CASP9, NOD1*, and *IL-18*, followed by in-group and qRT–PCR validation. Next, we divided all samples into high–low risk groups, based on the median of risk score. We enriched the differential genes between the two groups, compared the characteristics of immune infiltration and gene mutation between them, so as to explore the possible reasons for the influence of pyroptosis on prognosis. Finally, we combined the risk scores with other clinical features to create a prognostic model.

The caspases are proteases at the heart of networks that govern programmed cell death and play vital roles in the induction, transduction, and amplification of intracellular signals [28,29]. Caspases can be divided into three types: initiator, executioner, and inflammatory [30]. Caspase-6 is the executor caspase. It has been implicated in a number of neurological diseases and can be used as a therapeutic target [31]. However, the role of caspase-6 in pyroptosis remains unclear. Caspase-6 is not present in typical inflammasome activation and is not required for classic trigger-induced apoptosis and necrosis [32]. However, caspase-6 can cleave STAT1 to inhibit tumor cells in leukemia [33]. Similarly, STAT1 acts as a tumor suppressor in thyroid cancer. The long non-coding RNA transcriptionally mediated by STAT1 regulates thyroid cancer cell growth, migration, and invasion [34,35]. In the present model, *CASP6* was associated with a shorter DFS. Thus, it was speculated that a similar pathway exists in PTC recurrence. Inhibiting *CASP6* may be a new strategy to reduce PTC recurrence, but further basic studies are needed to verify this hypothesis. Different from caspase-6, caspase-9 is an initiator caspase, which is the initiation protein of the innate apoptosis pathway. Caspase-9 is activated by binding to Apaf-1 apoptotic bodies, triggering the CASPs cascade, which then activates CASP3 and leads to cell death [36]. Since it is located upstream of the entire pathway, caspase-9 is considered a promising therapeutic target. For example, previous studies have found that tumor cells can hijack caspase-9 signaling to suppress radiation-induced immunity [37]. The present study showed that *CASP9* was down-regulated in PTC and was a tumor suppressor gene. It might be involved in the process of killing tumor cells and can be inhibited in a tumor, although this mechanism remains unclear. In addition, the loss of *CASP9* signaling can lead to tumor immune escape and tumor recurrence by up-regulating PD-L1, which also provides new insights for immunotherapy of thyroid cancer.

NOD1 is one of the most important members of the NOD-like receptor family. NOD1 can detect conserved motifs in bacterial peptidoglycan and initiate nuclear factor-κB (NF-κB)-dependent and mitogen-activated protein kinase-dependent gene transcription, resulting in proinflammatory and antimicrobial responses [38,39]. Furthermore, *NOD1* has been reported to be involved in the development of many cancers, such as breast and colon cancer [40]. NF-κB is one of the most prominent cascades involved in tumor development [41], which regulates the proliferation and apoptosis signaling pathways of cancer cells. It has recently been shown to play an important role in thyroid cancer, with related genes, such as *RET/PTC, RAS*, and *BRAF*, acting through NF-κB [42,43]. However, the specific regulatory mechanisms of NOD1 and NF-κB in PTC remain unclear. IL-18 is a member of the IL-1 cytokine super family and is expressed in inflammation, autoimmune diseases, various cancers, and many infectious diseases [44]. Activation of caspase-1 and -11 can drive the release of IL-18 in pyroptosis [9]. In general, IL-18 plays a complex role in cancer. On one hand, it can stimulate NK and T cells, enhance the T helper type-1 immune response, and secrete interferon-γ, which effectively destroy cancer cells. On the other hand, IL-18 can induce angiogenesis, which helps tumor metastasis and proliferation. The creation of a chronic inflammatory microenvironment also helps tumor cells evade the immune response of host cells [45]. Therefore, *IL-18* may be considered to be a new tumor therapeutic target, and overcoming its avoidant immunophenotype may be a new cancer therapeutic strategy. In the present study, there was a difference in the expression of *IL-18* in the TCGA database and clinical samples, which might be related to the presence of inflammation in the patients themselves. In the risk score model, the higher the expression of *IL-18*, the longer the DFS, suggesting that *IL-18* might be involved in the process of inhibiting cancer cell growth, and increasing its expression could help improve the prognosis.

The relationship among the risk score, immune characteristics, and tumor mutation burden was then explored further. There is a complex network of interactions between the immune system and cancer, and understanding these interactions may improve immunotherapy treatment for thyroid cancer. The immune system can specifically destroy tumor cells and control tumor development through immune monitoring [46]. However, tumor cells can also reduce their immunogenicity by recruiting immunosuppressive cells and overexpressing immune checkpoints to achieve immune evasion [47]. Several previous studies, based on the TCGA data for thyroid cancer, have found that immune scores were associated with *BRAF* mutations, immune checkpoint expression, and prognosis [48,49]. In the present study, there were also differences between the high- and low-risk groups. Generally, the low-risk group showed a higher immune infiltration level and lower tumor purity. In terms of the immune checkpoint, previous studies suggested a high expression of PD1, PD-L1, and CTLA4 in PTC. In the present study, higher expression of CTLA4 was present in the low-risk group, while PD1 and PD-L1 showed no differences. In addition, higher *BRAF* mutation rates were evident in the low-risk group. BRAF inhibitors combined with immune checkpoint inhibitors (ICIs) were reported to significantly inhibit thyroid cancer growth and prolong survival in mouse models [50]. Therefore, these results suggested that patients in low-risk groups were more likely to benefit from ICI treatment. Finally, TMB differences between the two groups were compared. For tumors, the higher the TMB, the more neo-antigens were expressed and the more sensitive they were to ICIs [51]. For patients not treated with ICIs, a higher TMB was often associated with poor survival [52], while in patients treated with ICIs, a higher TMB was associated with longer survival [53]. The present study results were similar. TBM in the high-risk group was significantly higher than that in the low-risk group and was associated with TMB and risk score. In this regard, higher TMB offers new possibilities for immunotherapy in high-risk patients.

There were several limitations in the present study. First, due to the lack of DFS data, this research was based on a single database and lacked external validation. Second, the PRGs were obtained from previous literature. With the development of research on pyroptosis, there may be more PRGs that can be incorporated into the model in order to improve its accuracy. Third, few studies on the correlation between pyroptosis and thyroid cancer have been performed, and the role of pyroptosis in tumor development and immune characteristics needs to be confirmed by additional in vitro and in vivo studies.

## 5. Conclusions

In conclusion, the present study explored the role of PRGs in predicting PTC recurrence and treatment, constructed a gene signature, and discussed its relationship with immune characteristics. These results may contribute to risk stratification of PTC patients, helping to provide more personalized and precise treatments, in addition to new insights into the role of immunotherapy in PTC.

## Figures and Tables

**Figure 1 jpm-13-00085-f001:**
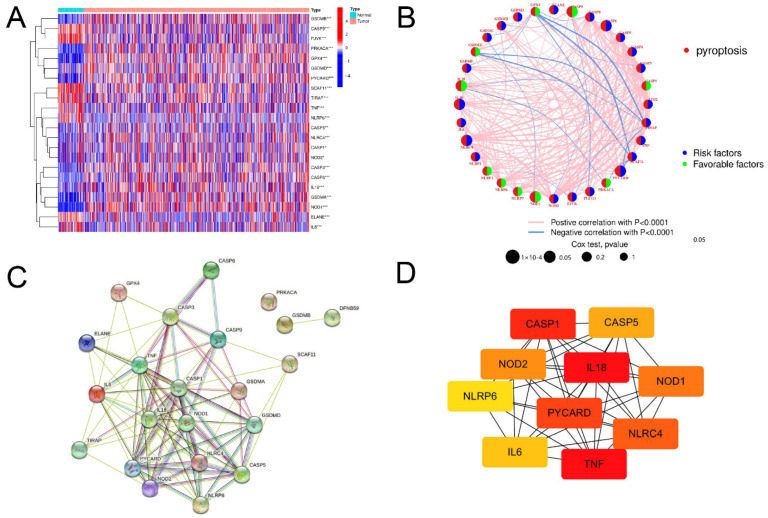
Identification of differentially expressed PRGs in normal and tumor samples. (**A**) Heatmap depicting variations in PRG expression between normal and tumor samples. * *p* < 0.05, ** *p* < 0.01, and *** *p* < 0.001. (**B**) RPG correlation network. (**C**) PPI network showing interactions of differentially expressed PRGs. (**D**) Network of top 10 hub genes. The color depth represents the ranking of hub genes. The darker the color, the higher the level of genes.

**Figure 2 jpm-13-00085-f002:**
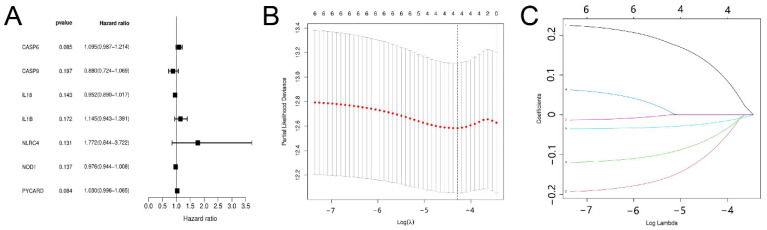
Univariate Cox regression and Lasso analyses. (**A**) PRGs with *p* < 0.2 were screened by univariate Cox regression. (**B**) Optimal parameter (λ) was chosen using cross-validation. (**C**) Lasso regression of six DFS-related PRGs.

**Figure 3 jpm-13-00085-f003:**
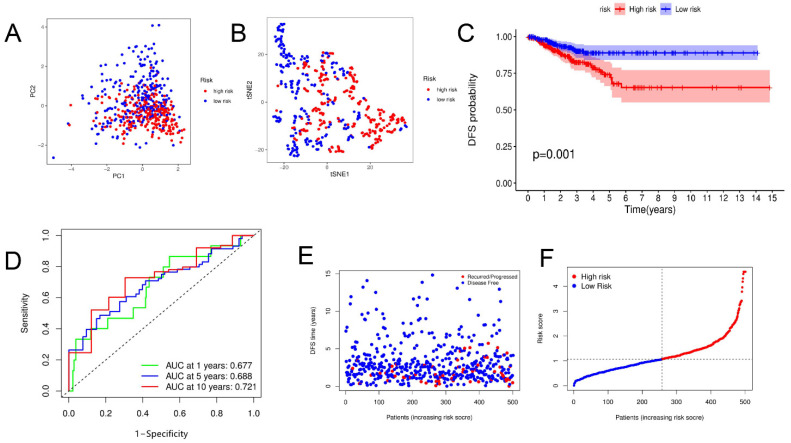
Construction of risk score in training set. (**A**,**B**) PCA and t-SNE plots based on risk score. (**C**) Kaplan–Meier curves for DFS in high- and low-risk cluster patients. (**D**) ROC curves demonstrating predictive efficiency of risk score. (**E**,**F**) Distribution of DFS status and risk score.

**Figure 4 jpm-13-00085-f004:**
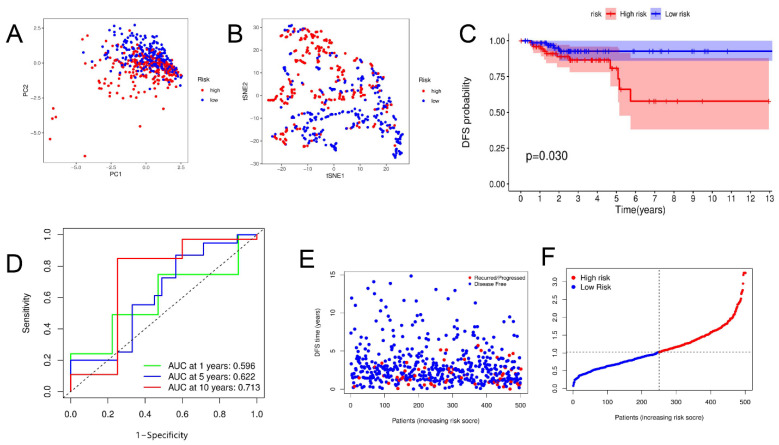
Construction of risk score in testing set. (**A**,**B**) PCA and t-SNE plots based on risk score. (**C**) Kaplan–Meier curves for DFS in high- and low-risk cluster patients. (**D**) ROC curves demonstrating predictive efficiency of risk score. (**E**,**F**) Distribution of DFS status and risk score.

**Figure 5 jpm-13-00085-f005:**
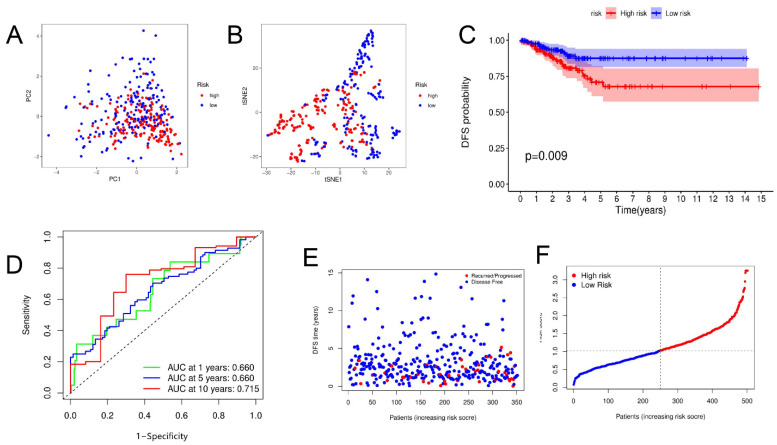
Construction of risk score in total set. (**A**,**B**) PCA and t-SNE plots based on risk score. (**C**) Kaplan–Meier curves for DFS in high- and low-risk cluster patients. (**D**) ROC curves demonstrating predictive efficiency of risk score. (**E**,**F**) Distribution of DFS status and risk score.

**Figure 6 jpm-13-00085-f006:**
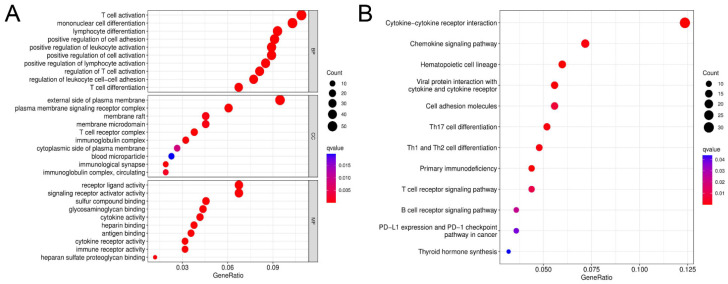
Functional enrichment analysis based on DEGs. (**A**) Bubble graph for GO enrichment. (**B**) Bubble graph for KEGG pathways. X-axis represents the proportion of the number of genes enriched to the target pathway in the total genes contained in the gene list. The size of the dots represents the number of genes enriched into this pathway, and different colors represent different adjusted *p*-values.

**Figure 7 jpm-13-00085-f007:**
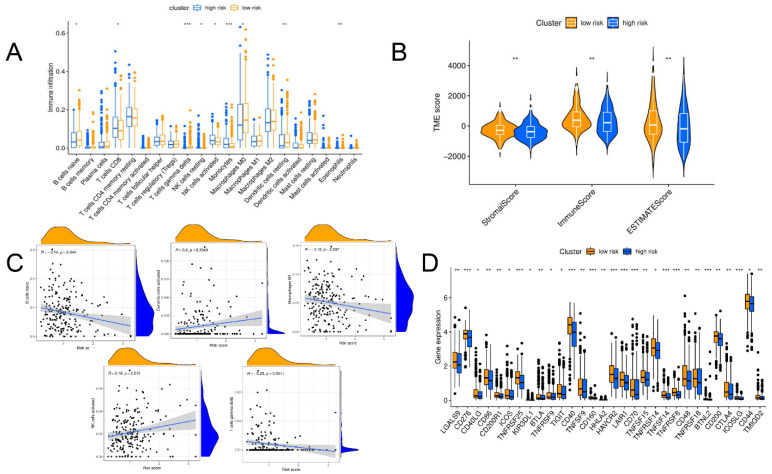
Immune characteristic analysis. (**A**) Comparison of infiltrating immune cell levels between high- and low-risk clusters. (**B**) Comparison of TME scores between high- and low-risk clusters. (**C**) Five immune cell types associated with risk score. (**D**) Differential expression of immune checkpoint genes between high- and low-risk clusters. * *p* < 0.05, ** *p* < 0.01, *** *p* < 0.001.

**Figure 8 jpm-13-00085-f008:**
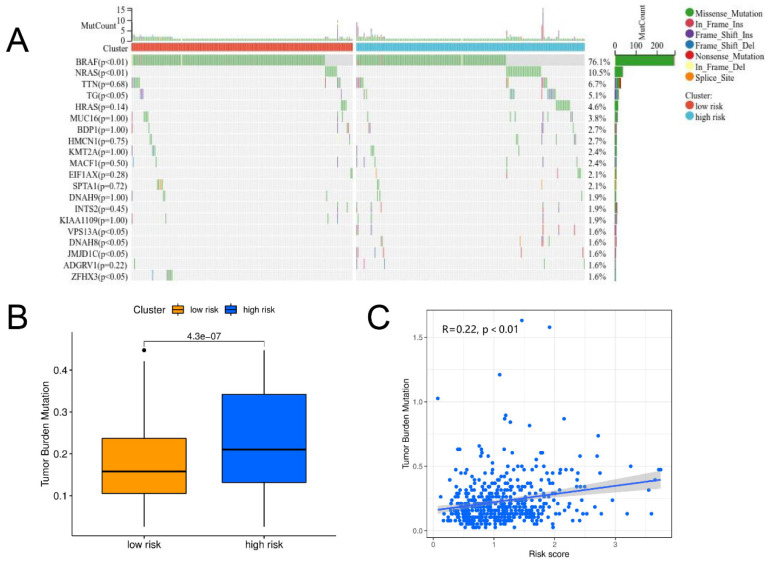
Gene mutation analysis. (**A**) Mutation profile and mutation rate comparison. (**B**) Comparison of TMB between high- and low-risk clusters. (**C**) Relationship between risk score and TMB.

**Figure 9 jpm-13-00085-f009:**
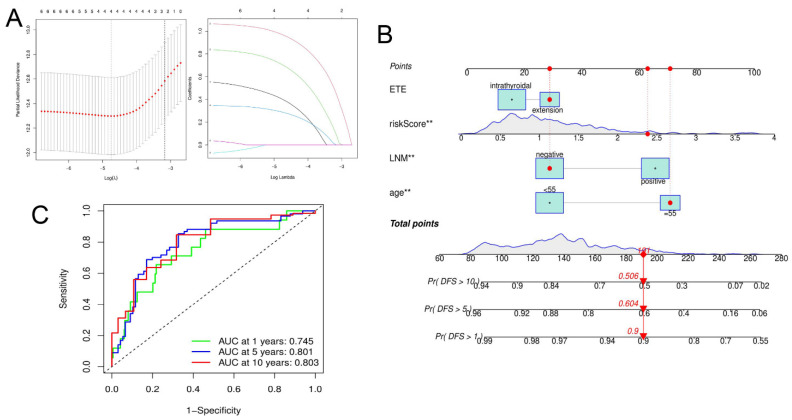
Construction and validation of nomogram to predict DFS. (**A**) Lasso regression was used to screen clinical factors. (**B**) The nomogram was built based on age, LNM, risk score, and ETE. (**C**) ROC curves demonstrated predictive efficiency of nomogram. ** *p* < 0.01.

**Figure 10 jpm-13-00085-f010:**
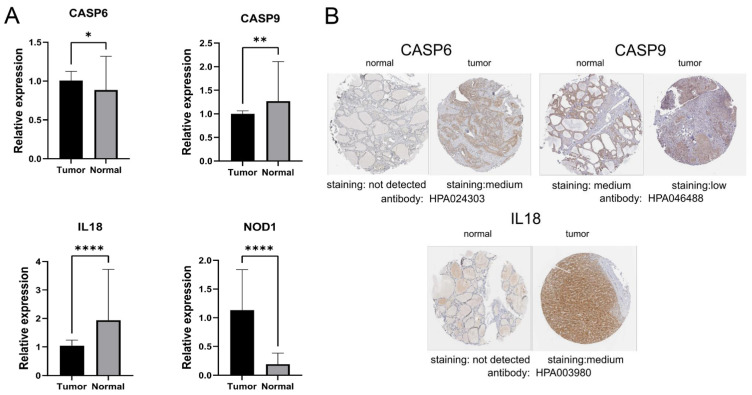
Expression of PRGs. (**A**) Relative expression of CASP6, CASP9, IL18, and NOD1 in thyroid tumor and normal tissue. (**B**) Immunohistochemistry analysis of CASP6, CASP9, and IL18 in normal and thyroid tumor tissue in HPA database. * *p* < 0.05, ** *p* < 0.01, **** *p* < 0.0001.

**Table 1 jpm-13-00085-t001:** Baseline clinical factors in TCGA patients.

	Total	Test	Train	*p* Value
CLNM				
CLNM(-)	203 (70.73%)	82 (71.93%)	121 (69.94%)	0.8184
CLNM(+)	84 (29.27%)	32 (28.07%)	52 (30.06%)	
Age				
<55	195 (67.94%)	75 (65.79%)	120 (69.36%)	0.6131
≥55	92 (32.06%)	39 (34.21%)	53 (30.64%)	
Sex				
Female	220 (76.66%)	93 (81.58%)	127 (73.41%)	0.1448
Male	67 (23.34%)	21 (18.42%)	46 (26.59%)	
Multifocality				
Multifocal	130 (45.3%)	49 (42.98%)	81 (46.82%)	0.6044
Unifocal	157 (54.7%)	65 (57.02%)	92 (53.18%)	
BRAF				
Mutant	152 (52.96%)	53 (46.49%)	99 (57.23%)	0.0965
Wild	135 (47.04%)	61 (53.51%)	74 (42.77%)	
ETE				
Extension	78 (27.18%)	26 (22.81%)	52 (30.06%)	0.2242
Intrathyroidal	209 (72.82%)	88 (77.19%)	121 (69.94%)	

## Data Availability

The data sets analyzed during this study are available in the TCGA database. (https://portal.gdc.cancer.gov/, 1 May 2022).

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
