# Peer review of "A Novel Pyroptosis-Related Gene Signature for Prediction of Disease-Free Survival in Papillary Thyroid Carcinoma"

_jpm, 2022, doi:10.3390/jpm13010085_

Round 1

Reviewer 1 Report

Jiang et al. successfully developed a Pyroptosis-related gene (PRG) score that predicts disease-free survival in papillary thyroid carcinoma (PTC).  Overall, the paper is very impressive and well done.  Unfortunately, however, there are several major issues and several minor issues that need to be addressed.

Major issues

 (1a) The pair-wise comparisons comparing the high-risk to low-risk clusters in Figure 7A, B, and D offer no formal statistical test.  Hotelling’s T2 analysis should be performed per set (see Rencher, 2002).  The multivariate test is to be preferred here (indeed insisted upon) because pairwise t-tests would incur the problem of multiple hypothesis testing.

(1b) On computing Hotelling’s T2, one should also use the more-focused methods in Rencher (2002) to determine the variable contributions to Hotelling’s T2 and their significance.  This will effectively by-pass the next major problem just following.

(2) The next major problem has to do with the correlations of the risk score and 5 selected immune cell types presented in Figure 7C.  There is no rigorous selection criterion given as to why these 5 cell types were selected.  Surely, their selection has to do with the results presented in Figure 7A, but, as mentioned in problem (2a) above, there was no formal test performed which would have shed light on the cell types for which the correlation with risk score should be investigated.

(3) In the text discussion of Figure 8, it was stated that “. . . there was a significant correlation between the risk score and TMB (Figure 8C).” However, examination of Figure 8C reveals that the quoted R (in the plot, that is) is 0.044 (which is very low), and the quoted p-value is 0.33, which is decidedly NOT significant.  Hopefully, this was a mislabeling and the quoted true value is indeed significant.  Otherwise, if the quoted values are correct, then one cannot say that the correlation between the risk score and TMB was significant.

Reference

Rencher AC. 2002.  Methods of Multivariate Analysis, (Wiley Series in Probability and Statistics).

Minor issues

(A) The AUCs quoted in Figures 3D, 4D, 5D, and 7C often do not match up with what is reported in the text.  Please ensure consistency between what is reported in the text and what is quoted on the figure.

(B) Please explain how to read and interpret the bubble graphs of Figure 6.

(C) The statistical test used for the comparison between tumor and normal for the 4 cell types should be stated.  Relatedly, indicate what the significance code is for Figure 10.  That is, what the p-values corresponding to the number of asterisks.

Author Response

Major issues

(1a) The pair-wise comparisons comparing the high-risk to low-risk clusters in Figure 7A, B, and D offer no formal statistical test.  Hotelling’s T2 analysis should be performed per set (see Rencher, 2002).  The multivariate test is to be preferred here (indeed insisted upon) because pairwise t-tests would incur the problem of multiple hypothesis testing.

(1b) On computing Hotelling’s T2, one should also use the more-focused methods in Rencher (2002) to determine the variable contributions to Hotelling’s T2 and their significance.  This will effectively by-pass the next major problem just following.

Response: Thank you for your suggestion. We fully agree that it is necessary to conduct multi-factor test between the high and low risk groups. We have conducted Hotelling's T2 test on the contents of Figure 7A, B and D respectively, and supplemented them in the manuscript.

Page:3, 8  Line number:114, 227-241

(1) Figure 7A. Comparison of immune cell infiltration in 22 groups between high and low risk groups.

We first tested the data for normality and homogeneity of variance, and the results showed that they met the requirements. Secondly, correlation test was conducted on the infiltration of these 22 kinds of immune cells, and the results also showed that there was correlation between them. Then, Hotelling's T2 test was further performed, and the result showed that F=3.772, P < 0.01, indicating that degree of immune infiltration in the low-risk group is significantly higher than that in the high-risk group. Hotelling's T2 test (P < 0.01) showsThe (Hotelling's T2, test P < 0.01) indicated that the level of immune cell infiltration in the low-risk group was higher than that in the high-risk group

(2) Figure 7B. Comparison of TME scores between high and low risk groups

Similarly, we conducted normality and variance homogeneity test on the data of the three scores, and the data met the requirements. Secondly, the correlation test shows that they are also correlated. Then, Hotelling's T2 test was further performed, and the results showed that F=4.568, P=0.011, suggesting that there were significant differences in the degree of immune infiltration between the two groups.

(3). Figure 7D. Comparison of immune checkpoint expression between high and low risk groups

Similarly, normality and variance homogeneity tests were conducted on the data of gene expression at immune checkpoint 47, and the data met the requirements. Secondly, the correlation test shows that they are also correlated. Then, Hotelling's T2 test was further performed, and the results showed that F=2.802, P < 0.01, indicating that there was a significant difference in the degree of immune infiltration between the two groups.

(2) The next major problem has to do with the correlations of the risk score and 5 selected immune cell types presented in Figure 7C.  There is no rigorous selection criterion given as to why these 5 cell types were selected.  Surely, their selection has to do with the results presented in Figure 7A, but, as mentioned in problem (1a) above, there was no formal test performed which would have shed light on the cell types for which the correlation with risk score should be investigated.

Response: Thanks for your kind comment. In Figure C, the method for selecting cells is to analyze the correlation between the infiltration degree of 22 kinds of immune cells and the risk score. The five immune cells as shown in the figure are significantly correlated with the risk score. And we added explanations to the manuscript.

Page:3  Line number:116

(3) In the text discussion of Figure 8, it was stated that “. . . there was a significant correlation between the risk score and TMB (Figure 8C).” However, examination of Figure 8C reveals that the quoted R (in the plot, that is) is 0.044 (which is very low), and the quoted p-value is 0.33, which is decidedly NOT significant.  Hopefully, this was a mislabeling and the quoted true value is indeed significant.  Otherwise, if the quoted values are correct, then one cannot say that the correlation between the risk score and TMB was significant.

Response: Thank you for your comments. We apologize for such an undeserved mistake. We examined and calculated the correlation coefficients and replotted the graph. The results indicate that the correlation coefficient R is 0.22 and P<0.01, indicating that there is a significant correlation between the two.

Page:11  Line number:258

Minor issues

  • The AUCs quoted in Figures 3D, 4D, 5D, and 7C often do not match up with what is reported in the text. Please ensure consistency between what is reported in the text and what is quoted on the figure.

Response: Thank you for your comments. We are sorry for such a mistake. We have carefully checked the data and revised them in the manuscript.

Page:5-6  Line number:184-193

(B) Please explain how to read and interpret the bubble graphs of Figure 6.

Response: Thank you for your comments. We have added the relevant explanations in the original text.

Page:8  Line number:221-224

In this part, we conducted GO and KEGG enrichment analysis of differential genes between high and low risk groups. GO enrichment analysis was further divided into three parts: Cellular component (CC), Biological process (BP) and Molecular function (MF). In the bubble diagram, the Y axis represents the cell's biological function and pathway; the X-axis represents the proportion of the number of genes enriched to the target pathway in the total genes contained in the gene list. The size of the dots represents the number of genes enriched into this pathway, and different colors represent different adjusted p-values. From blue to red, the enrichment degree of adjusted p-value becomes more and more significant from large to small. From blue to red, it means that the adjusted p-value increases from large to small, and the enrichment is more and more significant. As can be seen in Figure 7, the enrichment of differential genes was the highest in Biological process, especially in immune-related processes, including T cell activation, mononuclear cell differentiation, etc. In addition, the external side of plasma membrane, receptor ligand activity and other pathways also have a high degree of enrichment. In terms of KEGG enrichment, differential genes are mainly concentrated in the cytokine-cytokine receptor interaction and chemokine signaling pathway.

(C) The statistical test used for the comparison between tumor and normal for the 4 cell types should be stated. Relatedly, indicate what the significance code is for Figure 10. That is, what the p-values corresponding to the number of asterisks.

Response: Thank you very much for your comments. We have supplemented it in the manuscript and added the p value corresponding to the number of asterisks in the legend.

Page:4, 13  Line number:148-149, 288

For the qRT-PCR results, we used the paired T-test to compare tumor and adjacent tissue.

Reviewer 2 Report

(1) I found it interesting and important to apply machine learning methods to identify pyroptosis-related genes in papillary thyroid carcinoma patients and to help provide personalized and precise medicine. However, it is not easy to understand the Materials and Methods section as well as the associated figures with small fonts. Concise explanations for individual methods and their logical relationships are highly recommended.

(2) The abbreviation of TMB first appears on page 3 without the definition and that of TBM appears in the second last paragraph of the discussion, again without the definition. I suspect that TMB is short for "tumor mutation burden", although "tumor mutation load" and "Tumor Burden Mutation" are also confusingly used on page 8 and Figure 8B, respectively.  In addition, I assume that TBM and TMB are different, unless otherwise the sentence on page 13 does not make sense: TBM in the high-risk group ..... and was associated with TMB and risk score.

(3) The captions B and C  in Figure 2 appear to be reversed: Figure 2B indicates the optimal lambda whereas Figure 2C shows the lasso regression.

(4) Page 13: Finally, (r)isk score models were established based on four PRG(s), ...

Author Response

  1. I found it interesting and important to apply machine learning methods to identify pyroptosis-related genes in papillary thyroid carcinoma patients and to help provide personalized and precise medicine. However, it is not easy to understand the Materials and Methods section as well as the associated figures with small fonts. Concise explanations for individual methods and their logical relationships are highly recommended.

Response: Thank you for your advice. We are very sorry that we did not explain the method clearly. The overall method and logic of this paper is as follows. We first obtained the transcriptome data of 510 PTCS and 58 normal tissues from the TCGA database, and obtained 33 pyroptosis-related genes (PRGs) from the literature. Then we detected the expression of PRGs in PTC and normal tissues. Based on these differentially expressed genes, a risk score model for disease-free survival was established by LASSO regression and validated within the group. Next, we divided all samples into high-low risk groups based on the median of risk score. We enriched the differential genes between the two groups, compared the characteristics of immune infiltration and gene mutation between them, so as to explored the possible reasons for the influence of pyroptosis on prognosis. Finally, we combined the risk scores with other clinical features to create a prognostic model. In addition, we verified the expression of four PRGs in the scoring model in clinical tissue samples from our center.

We added the above explanation to the manuscript.

Page:13  Line number:296-305

2.The abbreviation of TMB first appears on page 3 without the definition and that of TBM appears in the second last paragraph of the discussion, again without the definition. I suspect that TMB is short for "tumor mutation burden", although "tumor mutation load" and "Tumor Burden Mutation" are also confusingly used on page 8 and Figure 8B, respectively.  In addition, I assume that TBM and TMB are different, unless otherwise the sentence on page 13 does not make sense: TBM in the high-risk group ..... and was associated with TMB and risk score.

Response: Thank you for pointing out our mistake. We corrected it in the manuscript.

Page:3, 9  Line number:122, 252-254

3.The captions B and C in Figure 2 appear to be reversed: Figure 2B indicates the optimal lambda whereas Figure 2C shows the lasso regression.

Response: Thank you for pointing out our mistake. We corrected it in the manuscript.

Page:6  Line number:196-197

4.Page 13: Finally, (r)isk score models were established based on four PRG(s), ...

Response: Thank you for pointing out our mistake. We corrected it in the manuscript.

Page:13  Line number:299

Reviewer 3 Report

Jiang et al. leverage the TCGA database to investigate the contribution of the expression of pyroptosis-related genes in predicting thyroid cancer progression and disease free-state. They find that utilizing four PRGs, they could form a model predictive of thyroid cancer recurrence. This model is further supported by experimental validation of gene expression, protein expression and tumor mutational burden.

The paper, while sound, bears close similarity to Pu Wu, Jinyuan Shi, Wei Sun & Hao Zhang's recent paper in Cancer Cell International (doi: 10.1186/s12935-021-02231-0). The two papers are similar in methodology and findings. The authors should cite this article in their paper andit would be good to discuss the differences between that paper and the research presented here.

In the discussion, the authors suggest a mechanism whereby Cas6 cleaves STAT1 to inhibit tumor cells in leukemia, however they later state that inhibition of Cas6 itself may be a therapeutic strategy. Therapeutic inhibition of Cas6 seemed counterintuitive to its aforementioned mechanism of inhibiting tumor cells, It would be beneficial for the authors to clarify this line of thinking.

There are no citations in the introduction or methodology. When citing R packages in the methodology, neither the R software nor version is mentioned. This should be added. Where possible, the authors should also reference papers relating to the R packages. This can be found on the documentation of the package, usually under "how to cite us". For example the citation for the limma package (according to their bioconductor page) is: Ritchie ME, Phipson B, Wu D, Hu Y, Law CW, Shi W, Smyth GK (2015). “limma powers differential expression analyses for RNA-sequencing and microarray studies.” Nucleic Acids Research, 43(7), e47. doi: 10.1093/nar/gkv007.

In Figure 1D genes are colored differently but there is no mention or explanation of what the different colors indicate. It would be clearer to add this key into the figure legend.

Author Response

1.The paper, while sound, bears close similarity to Pu Wu, Jinyuan Shi, Wei Sun & Hao Zhang's recent paper in Cancer Cell International (doi: 10.1186/s12935-021-02231-0). The two papers are similar in methodology and findings. The authors should cite this article in their paper and it would be good to discuss the differences between that paper and the research presented here.

Response: Thank you for your comments. We also noted this article. Indeed, both of these papers are based on TCGA database, and their research methods are similar. But the most obvious difference between the two is the end point of the study, one is OS and the other is DFS. As we have shown in this paper, for PTC, with standard treatment, the mortality rate and extremely low, 5-year survival rate can be more than 98%. Also taking the TCGA database as an example, only 16 of 510 PTC patients died, with a mortality rate of only 3%. And statistically speaking, there is also a large bias. In contrast, we pay more attention to the recurrence and progression of patients, which is also the trouble we encounter in clinical work. How to identify high and low risk patients who relapse, so as to provide more precise treatment and avoid overtreatment, may be more meaningful.

We added a quote and discussion of this papers to the manuscript.

Page:2  Line number:69-71

2.In the discussion, the authors suggest a mechanism whereby Cas6 cleaves STAT1 to inhibit tumor cells in leukemia, however they later state that inhibition of Cas6 itself may be a therapeutic strategy. Therapeutic inhibition of Cas6 seemed counterintuitive to its aforementioned mechanism of inhibiting tumor cells, It would be beneficial for the authors to clarify this line of thinking.

Response: Thank you for your comments. According to cited studies, Signal Transducer and Activator of Transcription-1 (STAT1) is overexpressed in some types of leukemia and is associated with poor prognosis. Caspase-6 can cleave STAT1 at multiple sites, thereby inhibiting STAT1, which may be beneficial. However, in other solid tumors, STAT1 is phosphorylated in response to interferon (IFN) stimulation, limiting cell proliferation and survival. In PTC, it has been reported that STAT1 activity is inhibited and associated with tumor characteristics such as tumor size and BRAF mutation. It has also been found that long non-coding RNA GAS5 inhibits the proliferation and metastasis of papillary thyroid carcinoma through IFN/STAT1 signaling pathway. In this study, we found that CASP6 was associated with poor prognosis. Based on this, we speculate that CASP6 can also cut STAT1 in PTC. It may be a therapeutic strategy to increase STAT1 expression by inhibiting CASP6.

3.There are no citations in the introduction or methodology. When citing R packages in the methodology, neither the R software nor version is mentioned. This should be added. Where possible, the authors should also reference papers relating to the R packages. This can be found on the documentation of the package, usually under "how to cite us". For example the citation for the limma package (according to their bioconductor page) is: Ritchie ME, Phipson B, Wu D, Hu Y, Law CW, Shi W, Smyth GK (2015). “limma powers differential expression analyses for RNA-sequencing and microarray studies.” Nucleic Acids Research, 43(7), e47. doi: 10.1093/nar/gkv007.

Response: Thank you for your comments. We are very sorry for the omission. R software and its versions are mentioned in the methods section 2.9. Then, all the R packages used in the article are supplemented with references.

4.In Figure 1D genes are colored differently but there is no mention or explanation of what the different colors indicate. It would be clearer to add this key into the figure legend.

Response: Thank you for your suggestion. We think it is very meaningful. In Figure 1D, the color depth represents the ranking of hub genes. The darker the color, the higher the level of genes. In addition, we have supplemented relevant descriptions in the legend.

Page:5  Line number:170-171

Reviewer 4 Report

This work is devoted to the study of the gene signature, which will allow to personalize the risks of recurrence of papillary thyroid carcinoma. The results obtained, although partially contradicting the data obtained on gene expression, are undoubtedly important. The accumulation of data from various groups of researchers will allow further meta-analysis. Since all the identified target genes are involved in one pathogenesis pathway, I would advise the authors to make calculations based on a combination of data on the expression of these genes against a combination in the control.

Author Response

Thank you very much for your comments, we think your advice is very meaningful. We used the risk scoring model established in this study to score 58 normal thyroid tissues in the control group, and the results are shown in the figure1. Then we used the unpaired Students T-test to compare tumor and normal tissue scores. The results are shown in the figure2, however, there was no statistical difference between the two groups. This suggests that the risk score model established in this paper may not be suitable for the diagnosis of thyroid cancer, and there are differences between the diagnostic model and the prognostic model. In addition, by examining the expression levels of pyroptosis-related genes in tumor and normal tissues, we found that three genes associated with better prognosis were downregulated in tumor tissues. Due to the bidirectional effect of pyroptosis in cancer, we hypothesized that pyroptosis plays a different role in the two processes of tumor occurrence and recurrence. However, there are still few studies on the role of pyroptosis in the development and progression of thyroid cancer, and we expect more studies to focus on this in the future.

figure 1

figure 2

Round 2

Reviewer 3 Report

The authors have responded to all of my comments and I am satisified with the changes and/or explanations provided. I thank the authors for taking the time to address these.